# A short-lived peptide signal regulates cell-to-cell communication in *Listeria monocytogenes*
Benjamin S. Bejder [1], Fabrizio Monda [1,3], Bengt H. Gless [1,4], Martin S. Bojer [2], Hanne Ingmer[2] & Christian A. Olsen [1] ✉

Quorum sensing (QS) is a mechanism that regulates group behavior in bacteria, and in Gram-positive bacteria, the communication molecules are often cyclic peptides, called autoinducing peptides (AIPs). We recently showed that pentameric thiolactone-containing AIPs from *Listeria monocytogenes*, and from other species, spontaneously undergo rapid rearrangement to homodetic cyclopeptides, which hampers our ability to study the activity of these short-lived compounds. Here, we developed chemically modified analogues that closely mimic the native AIPs while remaining structurally intact, by introducing N-methylation or thioester-to-thioether substitutions. The stabilized AIP analogues exhibit strong QS agonism in *L. monocytogenes* and allow structure–activity relationships to be studied. Our data provide evidence to suggest that the most potent AIP is in fact the very short-lived thiolactone-containing pentamer. Further, we find that the QS system in *L. monocytogenes* is more promiscuous with respect to the structural diversity allowed for agonistic AIPs than reported for the more extensively studied QS systems in *Staphylococcus aureus* and *Staphylococcus epidermidis*. The developed compounds will be important for uncovering the biology of *L. monocytogenes*, and the design principles should be broadly applicable to the study of AIPs in other species.

*Listeria monocytogenes* is a notorious foodborne pathogenic Gram-positive bacterium, causing intracellular infections in humans and animals. It is found in soil and wastewater, and can persist in food-processing facilities, even after sanitization. When ingested, *L. monocytogenes* can cause disease ranging from gastroenteritis to life-threatening conditions in immuno-compromised, pregnant, or elderly individuals. Being able to survive a wide range of environmental conditions requires tightly regulated gene expression[1,2]. This is in part accomplished by quorum sensing (QS)[3–9] that in *L. monocytogenes* affects protein secretion, cell invasion, biofilm formation, and virulence gene expression. QS constitutes regulatory systems for sensing population density, through excretion and recognition of autoinducer molecules, causing changes in population behavior in different bacteria[10]. In Gram-positive bacteria, the best characterized QS system is that encoded by the accessory gene regulator (*agr*) locus in *Staphylococcus aureus*[11]. In this system, the signaling molecules are cyclic, autoinducing peptides (AIPs), that commonly contain a thiolactone formed between the C-terminal carboxylate and the thiol side chain of a cysteine residue in the *i*–4 position

from the C-terminus, together with a linear N-terminal exotail of varying length[12–16]. The *agr* locus encodes the AIP precursor peptide (AgrD) and a membrane-bound peptidase (AgrB) needed for the closing of the thiolactone ring. Additionally, it encodes a two-component system, consisting of the receptor-histidine kinase (AgrC) and response regulator (AgrA) that together comprise the AIP-sensing and signal transduction components[17]. The *agr* system of *L. monocytogenes*, like that of *S. aureus*, contains the four genes, *agrBDCA*, that are autoregulated and expressed from the same promoter[3,4]. The system has been known for 20 years and has been shown to affect the expression of hundreds of genes[6,18], but its regulatory function is not yet well understood.

The identity of the *L. monocytogenes* AIP is debated. Previously, a hexameric AIP (**P1**) with a single amino acid exotail was identified using LC-MS/MS[19,20], and Zetzmann et al.[8] provided evidence to suggest a pentameric AIP (**P2**) without an exotail (Fig. 1). Previously, we failed to identify **P1** by using a thiolactone-trapping methodology[16,21] and showed that AIPs without an exotail spontaneously undergo S→N acyl shift to give homodetic

[1]Center for Biopharmaceuticals and Department of Drug Design and Pharmacology, Faculty of Health and Medical Sciences, University of Copenhagen, Copenhagen, Denmark. [2]Department of Veterinary and Animal Sciences, Faculty of Health and Medical Sciences, University of Copenhagen, Frederiksberg C, Denmark. [3]Present address: Nuevolution A/S, Amgen Research Copenhagen, Copenhagen, Denmark. [4]Present address: Yusuf Hamied Department of Chemistry, University of Cambridge, Cambridge, UK. ✉e-mail: cao@sund.ku.dk

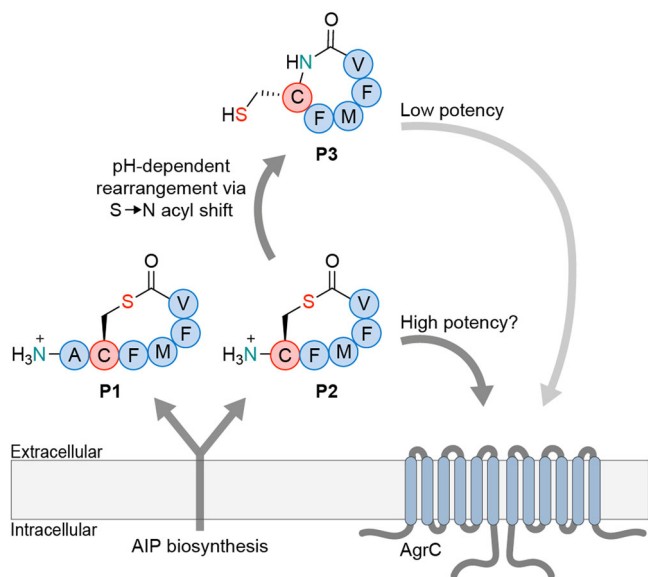

**Fig. 1 | Cartoon representation of the *agr* system in *Listeria monocytogenes*.** Structures of the proposed autoinducing peptides **P1** and **P2** as well as the homodetic peptide **P3**, which is spontaneously formed by rearrangement of **P2**, are shown.

peptides, like **P3** (Fig. 1)[21]. Further, we were able to identify **P3** in supernatant from *L. monocytogenes* bacterial culture and showed that it can induce QS in a bioluminescent reporter strain of *L. monocytogenes*; albeit, at high concentrations (EC$_{50}$ ~ 8 µM) compared to those needed for autoinduction in staphylococci[16]. Concurrently, a similar S→N acyl shift phenomenon was described in the bacterium *Ruminiclostridium cellulolyticum*[22]. Although our work demonstrated the rapid rearrangement of **P2** to give **P3** at pH 7, the high concentration needed for **P3** to activate QS still led us to speculate whether the **P2** thiolactone could play a role in the activation of QS in *L. monocytogenes*.

Here, we report the development of stabilized analogues, mimicking the structure of **P2**, to investigate whether **P2** could act as a short-lived but highly potent activator of the *L. monocytogenes* QS system.

## Results

### Assay optimization and evaluation of compounds P1–P3

First, we assessed the rate of rearrangement from **P2** to **P3** at 37 °C, which is the relevant temperature when the bacteria infect humans, and here **P2** rearranged with a half-life of 1.3 min at pH 7 (Fig. 2a). For *S. aureus*, the AIP concentration in cultures at early stationary phase have been determined to be ~1 µM[23], and if assuming a similar concentration of **P2**, the remaining concentration of thiolactone-containing peptide after 9 and 13 min would be <10 nM and <1 nM, respectively (Fig. 2b). Albeit it would be difficult to imagine building up a concentration as high as 1 µM for **P2**, because of its rapid, continuous degradation. Nevertheless, this exercise suggests that if **P2** were to act as signaling molecule in a cell density sensing system, it would require rapid receptor activation at low peptide concentrations.

To address this question, we compared the effects of **P1**–**P3** on *agr*-dependent bacterial reporter strains at 37 °C, using *L. monocytogenes* EGDe WT and a Δ*agrD* mutant strain, both carrying a chromosomal integration of the *agr* promoter fused to the *lux* operon, enabling a bioluminescent readout as measure of *agr* activity (referred to as WT::P2-lux and Δ*agrD*::P2-lux, respectively)[8]. To improve signal-to-noise we first optimized the assay conditions inspired by a study on luciferase-based reporter strains of *Lactococcus lactis*, in which the luciferase substrate flavin mononucleotide was added[24]. We found that addition of flavin mononucleotide (10 mg/L, ~26 µM) amplified maximal luminescence readout of the WT::P2-lux reporter strain by ~80% and total area under the curve by ~90% when grown in tryptic soy broth (TSB), without affecting the growth (Supplementary Fig. 1). We then

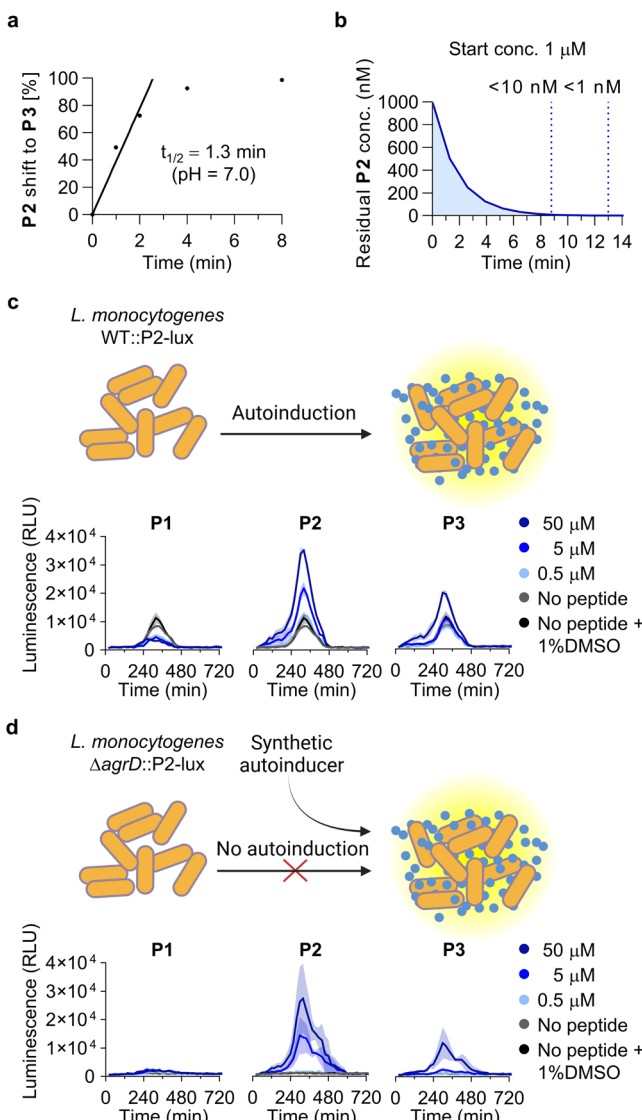

**Fig. 2 | Rearrangement of P2 and activity of P1–P3. a** UPLC-based assessment of the rearrangement rate of **P2** to give **P3**. **b** Plot of the decaying concentration of **P2** from a starting concentration of 1 µM (based on the determined $t_{1/2}$ = 1.3 min). **c** Effect of selected concentrations of **P1**–**P3** on the *agr* activity in luminescence-based reporter strain WT::P2-lux of *L. monocytogenes* grown at 37 °C in tryptic soy broth (TSB) medium. **d** Effect of selected concentrations of **P1**–**P3** on the *agr* activity in luminescence-based reporter strain Δ*agrD*::P2-lux of *L. monocytogenes* grown at 37 °C in tryptic soy broth (TSB) medium. Shadings on the graphs represent the standard error of the mean (SEM). All reporter strain assay data is based on three individual assays (n = 3) performed in at least technical duplicate. "No peptide" control wells contain pure TSB medium corresponding to the volume added for the dilution series of the peptide. "No peptide +1% DMSO" control wells contain 1% DMSO, corresponding to the concentration of DMSO at the highest concentration of synthetic peptide (50 µM). Part of the figure was prepared using BioRender. Figure adapted from the PhD thesis of B. S. Bejder entitled "*Chemical Microbiology Investigations of Peptide-based Cell-to-Cell Communication in Gram-positive Bacteria*" (University of Copenhagen, 2024).

tested synthetic peptides **P1**–**P3** against the WT::P2-lux and Δ*agrD*::P2-lux reporter strains in TSB and brain heart infusion (BHI) medium at 37 °C. Importantly, **P2** was added directly from DMSO stock solution, to preserve the integrity of the thiolactone motif for as long as possible during the experiment (Fig. 2c, d and Supplementary Figs. 2–5). The trends previously observed for **P1** and **P3** at 30 °C were recapitulated under the optimized conditions. Treating WT::P2-lux with **P1** resulted

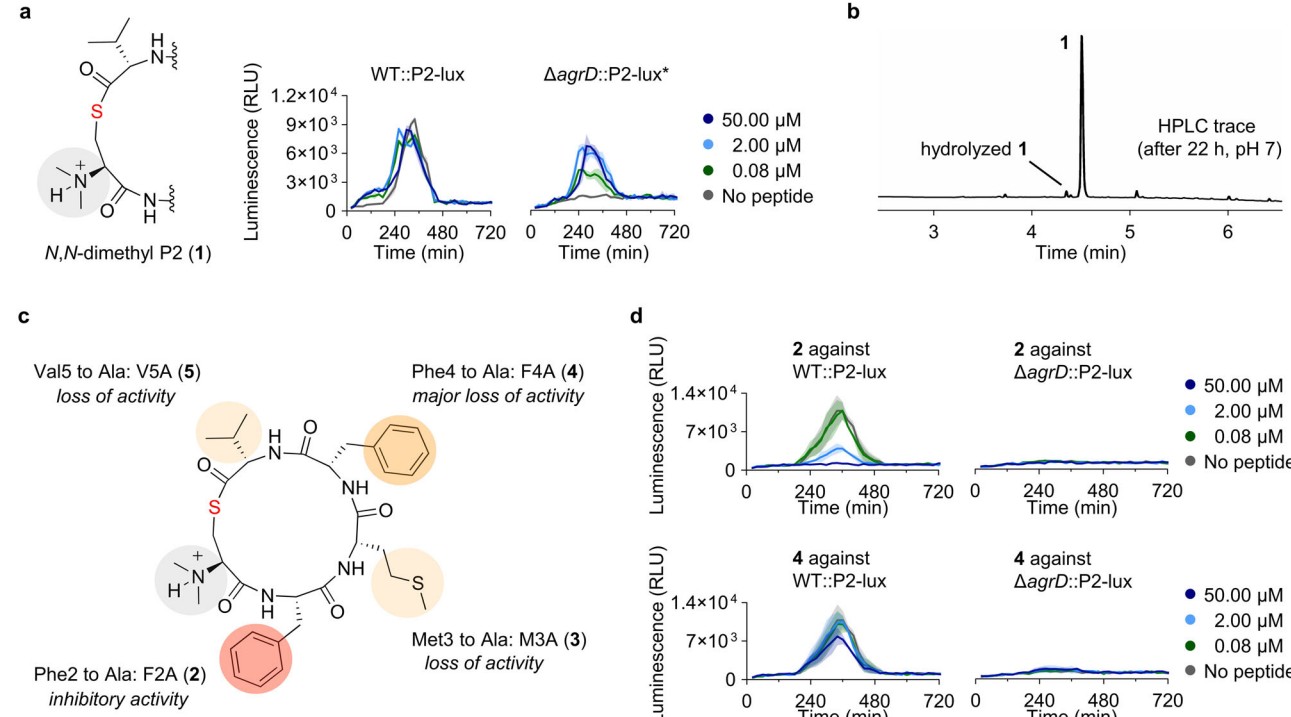

**Fig. 3 | Chemical structures of peptide analogues 1–5, rearrangement evaluation of 1, and selected reporter strain assay data. a** Structural modification to give **1** and effect of selected concentrations of **1** on the *agr* activity in luminescence-based reporter strains of *L. monocytogenes* grown at 37 °C in tryptic soy broth (TSB) medium. *Based on two individual assays (n = 2) performed in technical triplicate. **b** HPLC analysis of the stability of **1** after incubation for 22 h in neutral buffer. **c** Overview of the effects of alanine substitution of compound **1**. **d** Effects of selected concentrations of **2** and **4** on the *agr* activity in luminescence-based reporter strains of *L. monocytogenes* grown at 37 °C in tryptic soy broth (TSB) medium, representing three individual assays (n = 3) performed in at least technical duplicate. "No peptide" control wells contain pure TSB medium corresponding to the volume added for the dilution series of the peptide. Shadings on the graphs represent the standard error of the mean (SEM).

## Table 1 | Activation of *agr* in *L. monocytogenes*[a]

| compound | %-activation [2 μM] | %-activation [80 nM] |
|---|---|---|
| P1 | 11 ± 6[b] | inactive[c,d] |
| P2 | **160 ± 51**[b] | inactive[c,d] |
| P3 | 15 ± 6[b] | inactive[c,d] |
| 1 | 63 ± 5 | 35 ± 3 |
| 6 | 27 ± 3 | inactive[c] |
| 7 | **210 ± 10** | 38 ± 2 |
| 8 | **270 ± 10** | 43 ± 8 |
| 9 | 80 ± 5 | 74 ± 8 |
| 10 | **350 ± 12** | **274 ± 7** |

Activation levels surpassing 100% are highlighted in bold.

[a]Values are extracted from assays with Δ*agrD*::P2-lux and normalized to average maximum luminescence of untreated Δ*agrD*::P2-lux (0%) and average maximum luminescence of untreated WT::P2-lux (100%).
[b]Tested at 5 μM.
[c]Inactive defined as less than 10% normalized activation.
[d]Tested at 500 nM.

in inhibition of the signal in both BHI and TSB media, while giving rise to a slight induction in the Δ*agrD*::P2-lux reporter strain after 240 min, which was most pronounced in BHI medium (Fig. 2c and Supplementary Figs. 2–5). The homodetic peptide **P3** did not cause inhibition of WT::P2-lux and produced early induction of *agr* in both TSB and BHI media (Fig. 2c, Supplementary Figs. 2 and 4). More interestingly, **P2** was able to increase *agr* activity 3-fold at 50 μM and 2-fold at 5 μM for WT::P2-lux in TSB, compared to the untreated WT::P2-lux (Fig. 2c, Supplementary Figs. 2 and 4).

A less pronounced increase in peak intensity was also observed for **P2** at 5 μM in BHI (Supplementary Fig. 4) and early induction of *agr* was observed in both TSB and BHI (Fig. 2c and Supplementary Figs. 2 and 4). For the activation of Δ*agrD*::P2-lux by **P2**, we observed full restoring of WT levels in BHI and surpassing the WT level in TSB, even at 5 μM compound concentration (Fig. 2d and Supplementary Figs. 3 and 5; see the Supplementary Information page 8 for further discussion). Even though **P2** is inherently unstable at near-neutral pH, its ability to activate *agr* at lower concentrations than **P1** and **P3** supports its function as a bona fide AIP of *L. monocytogenes*, but its potency remains elusive.

### Development of N-terminally modified mimics of P2

Previously, we synthesized an *N*-acetylated version of **P2**, to obtain a stable thiolactone mimic of **P2**, but this peptide showed no activation of Δ*agrD*::P2-lux and inhibited the *agr* activity of WT::P2-lux[21], which we confirmed with the optimized assay protocol (Supplementary Fig. 6). Since the *N*-acetylated mimic cannot be protonated, we speculated if a positively charged amino group at the N-terminus, as present in **P2**, could be important for activity. We therefore designed a stabilized mimic of the compound by introducing two methyl groups instead of acetyl (**1**; Fig. 3a and Supplementary Fig. 7). The two methyl groups prevented S→N acyl shift as intended, and we observed >90% purity of compound **1** after incubation for 22 h in phosphate buffer (pH 7.0) at 37 °C (Fig. 3b and Supplementary Fig. 8).

Remarkably, the thiolactone **1** caused early activation of *agr* in the WT::P2-lux strain and appreciable induction of *agr* in Δ*agrD*::P2-lux at just 80 nM compound concentration, displaying superior potency to any of the previously tested peptides (Fig. 3a, Table 1, Supplementary Fig. 9). However, thiolactone **1** was unable to increase *agr* activity above the WT levels, unlike the effect observed with **P2** and WT::P2-lux. Similarly, against Δ*agrD*::P2-

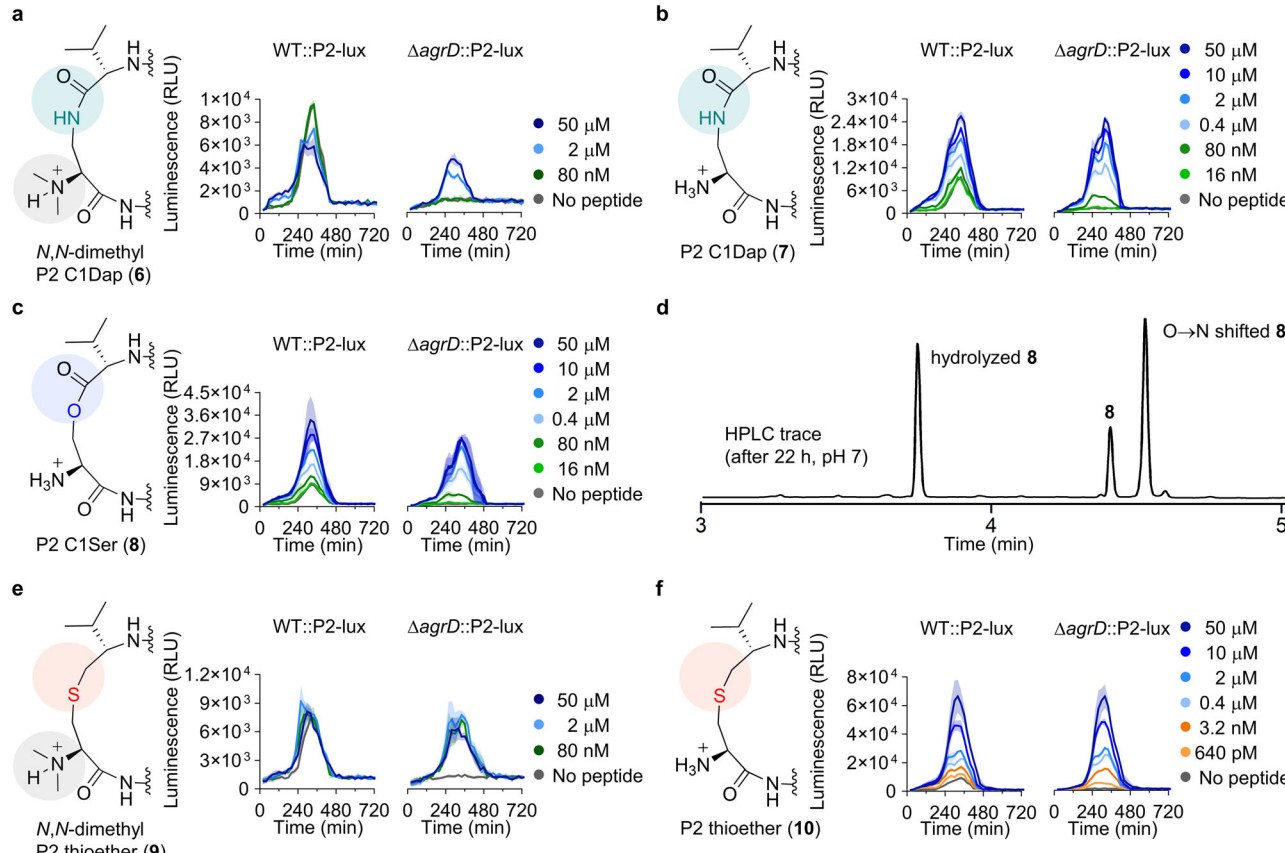

**Fig. 4 | Chemical structures of peptide analogues 6–10, rearrangement evaluation of 8, and selected reporter strain assay data. a–c** Structural modifications to give **6–8** and effect of selected concentrations of **6–8** on the *agr* activity in the reporter strains of *L. monocytogenes*. **d** HPLC trace showing the degree of rearrangement and hydrolysis of **8** after incubation in neutral buffer for 22 h. **e, f** Structural modifications to give **9** and **10** and effect of selected concentrations of **9** and **10** on the *agr* activity in the reporter strains of *L. monocytogenes*. Data is based on assays performed at 37 °C in tryptic soy broth (TSB) medium, representing three individual assays (n = 3) performed in at least technical duplicate. "No peptide" control wells contain pure TSB medium corresponding to the volume added for the dilution series of the peptide. Shadings on the graphs represent the standard error of the mean (SEM).

lux we observed an upper limit of activation (~6500 RLU) with this **P2** analogue, which was lower than the maximum luminescence of the WT::P2-lux strain (Fig. 3a and Supplementary Fig. 10). Nevertheless, the potency of **1** supports the hypothesis that *N*-acetylation is inappropriate for studying the biological activity of exotail-free thiolactone-containing AIPs. Dimethylation of the N-terminus therefore provides another architecture to achieve stabilization of these AIPs (Fig. 3b and Supplementary Fig. 8).

Having a potent **P2** mimic in hand, we performed an alanine scan based on **1** (peptides **2–5**; Fig. 3c), to study the effect of each position in **P2** on *agr* activity. We observed the most pronounced effect when substituting either phenylalanine (**2** and **4**), which resulted in a complete loss of activation of the Δ*agrD*::P2-lux reporter (Fig. 3d and Supplementary Figs. 9 and 10). This is in agreement with results reported for analogues of **P1** tested in a fluorescence reporter assay[20]. Interestingly, compound **2** was even able to inhibit *agr* activity in the WT::P2-lux strain at 2 and 50 µM concentration (Fig. 3d), providing an avenue for development of antagonists of the *L. monocytogenes agr* system that can function as tool compounds, which was also investigated in the recent study of **P1** analogues[20]. In contrast to **2** and **4**, the alanine analogues **3** and **5** were partial agonists unable to reach the same maximum activation of Δ*agrD*::P2-lux as **1**. Compounds **3** and **5** also displayed inhibition of peak luminescence in the WT::P2-lux, with analogue **5** being able to inhibit the activity by 20% at just 80 nM concentration (Supplementary Figs. 9 and 10). Although all alanine substitutions affected the agonistic effect, only modest inhibitors were discovered.

## Development of alternative stabilized analogues of P2

To explore alternative chemistry to provide stabilized AIP analogues, we synthesized lactam analogues of **1** and **P2** (**6** and **7**, respectively; Fig. 4a, b and Supplementary Figs. 11 and 12). The lactam analogue **7** proved to be an effective agonist that increased *agr* activity of WT::P2-lux above controls at concentrations down to 80 nM (Fig. 4b, Table 1, and Supplementary Fig. 13). At the highest concentration tested for **7** (50 µM), the culture reached the same maximal level of *agr* activity in WT::P2-lux and Δ*agrD*::P2-lux, around 2.5-fold higher than the untreated WT control (Fig. 4b and Supplementary Fig. 13), which is slightly lower than the maximal activation achieved with the native, short-lived compound **P2**. With compound **6** we observed an upper maximal *agr* activation in Δ*agrD*::P2-lux similar to the effect observed for **1** and, contrary to compound **7**, this analogue did not induce *agr* at 80 nM concentration (Fig. 4a, Table 1, and Supplementary Fig. 13). For comparison, structure-activity relationship studies performed on AIPs from *S. aureus*, demonstrated that lactam analogues of AIP-I and AIP-II were also still agonists of their cognate AgrC receptor; although, with ~1000-fold decrease in potency[25–27].

We further synthesized a lactone analogue of **P2** (**8**; Fig. 4c and Supplementary Fig. 14), which showed a similar degree of induction and potency as the lactam **7**, reaching roughly the same maximal luminescence levels in Δ*agrD*::P2-lux and half-maximal activation at 400 nM (Fig. 4c and Supplementary Fig. 15c). However, the lactone also proved inherently unstable, which causes an underestimation of its activity, as discussed for compound **P2**. Thus, after 22 h at 37 °C and pH 7.0, 36% of **8** was

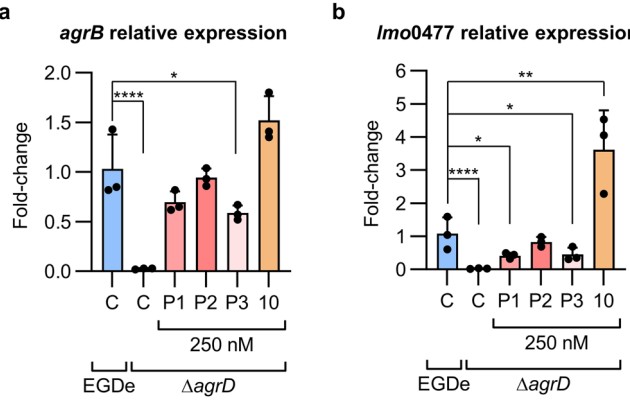

**Fig. 5 | Effect of synthetic AIP variants on *agr*-regulated gene expression assessed by quantitative real-time PCR (qPCR).** Relative quantification of *agrB* (**a**) or *lmo0477* (**b**) expression at late exponential growth phase (6 h) in *L. monocytogenes* EGDeΔ*agrD* grown with or without selected synthetic peptides [**P1, P2, P3**, or **10** (all added at 250 nM final concentration)] and compared with expression levels in *L. monocytogenes* EGDe WT. Performed in biological triplicate (n = 3). Error bars represent standard deviation. Statistical significance was evaluated by a one-way ANOVA: * ($p < 0.05$), ** ($p < 0.005$), *** ($p < 0.005$), **** ($p < 0.0001$).

hydrolyzed and 48% had undergone rearrangement through O→N acyl shift, to form its corresponding homodetic peptide (Fig. 4d and Supplementary Fig. 15a, b). Earlier work on *S. aureus* AIP-II showed that a lactone analogue of the AIP did not display agonist activity against its cognate receptor at concentrations <5 μM[25].

Finally, we extended our series to include thioether analogues **9** and **10** (Fig. 4e, f and Supplementary Figs. 16, 17). When treating WT::P2-lux with the analogue that bears the most structural resemblance to **P2**–compound **10**–we observed a striking 8-fold increase in *agr* activation compared to untreated WT. Compound **10** provided early induction in WT::P2-lux and increased *agr* activity of Δ*agrD*::P2-lux to ~50% of the untreated WT levels at concentrations as low as 640 pM (Fig. 4f and Supplementary Fig. 18). Furthermore, at just 3.2 nM compound concentration, the luminescence levels observed for Δ*agrD*::P2-lux paralleled those of untreated WT::P2-lux (Fig. 4f and Supplementary Fig. 18). For Δ*agrD*::P2-lux treated with compound **9**, we observed attenuated maximal activation, as for *N,N*-dimethylated analogues **1** and **6** (Fig. 4e and Supplementary Fig. 18). However, maximal induction of *agr* was observed at 80 nM concentration of **9**, rendering this compound a more potent mimic of the native AIP than the lactam **6** (Table 1).

### Reporter-independent assessment of P1–P3 and 10 on QS and downstream gene expression

Because of the differing results obtained for some compounds between the experiments performed with fluorescent reporter strains[20] and the luciferase reporter strains applied herein, we decided to investigate the QS activation at the RNA level to circumvent potential interference from an artificial reporter system. To this end, we tested the effect of 250 nM of selected peptides (**P1, P2, P3**, and **10**) on *agrB* expression in late exponential growth phase (6 h) in the parental strains of the luciferase reporter strains by quantitative real-time PCR (qPCR) (Fig. 5 and Supplementary Fig. 19a). As expected, we observed a significantly lower expression (~38-fold, $p < 0.0001$) of *agrB* in the Δ*agrD* mutant of *L. monocytogenes* EGDe, compared with the WT (Fig. 5a).

While the peptides **P1–P3** could not activate the Δ*agrD*::P2-lux reporter strain at 500 nM (Fig. 2d), we found that all three peptides could significantly increase *agrB* expression in the parental strain at 250 nM peptide concentration. At this concentration, both **P1** and **P2** could restore *agrB* expression in the Δ*agrD* mutant to levels that were not significantly different from the untreated WT (Fig. 5a) [67% of WT *agrB* expression with **P1** ($p = 0.08$) and 91% with **P2** ($p = 0.99$)]. The thioether analogue **10**

increased *agrB* expression in the Δ*agrD* mutant to 147% compared to the untreated WT level, however, this difference was not statistically significant ($p = 0.06$) (Fig. 5a). None of the peptides caused a significant change in *agrB* expression in the WT strain at 250 nM concentration (Supplementary Fig. 19a).

Further, we performed qPCR to assess the expression of *lmo0477*, a putative secreted protein that has previously been identified among the most downregulated genes in a Δ*agrA* mutant of *L. monocytogenes* EGDe at both 25 and 37 °C[18,28]. In correspondence with previous findings in a Δ*agrA* mutant[18], we could demonstrate that *lmo0477* was also downregulated by 36-fold in the Δ*agrD* mutant compared to the WT strain ($p < 0.0001$) (Fig. 5b). Strikingly, when treating the Δ*agrD* mutant with 250 nM of either **P1, P2**, or **P3**, only **P2** could restore *lmo0477* expression to a level that was not significantly different from the untreated WT strain ($p = 0.94$) (Fig. 5b). Moreover, the thioether analogue of **P2** (compound **10**) caused an increase in the expression of *lmo0477* in the Δ*agrD* mutant of ~3.6-fold beyond the level of the untreated WT strain ($p = 0.0037$) (Fig. 5b). Again, none of the peptides significantly affected the *lmo0477* expression levels in the WT at 250 nM (Supplementary Fig. 19b), as also observed for *agrB* expression.

## Discussion

In summary, we first investigated the mimicking of exotail-free thiolactone-containing AIPs by incorporating methyl groups at the N-terminus of **P2** to circumvent its S→N acyl shift to give **P3**. This strategy for studying exotail-free AIPs enabled us to investigate the importance of each amino acid in the peptide through an alanine scan. While all positions are indispensable for full activity, the phenylalanine positions proved most important for retaining *agr* activation.

Unlike what is known from studies of the paradigm *agr* systems in *S. aureus*, agonistic activity was retained when the native thiolactone functionality was substituted for a lactam (**7**) or a lactone (**8**) motif (Table 1). Moreover, an unprecedented thiolactone-to-thioether substitution, to give compound **10**, yielded the most potent inducer of the *L. monocytogenes agr* system reported so far, to the best of our knowledge (Table 1, Fig. 4f, and Supplementary Fig. 18). Comparison of lactam and thioether analogues with and without dimethylated N-termini (**6** vs **7** and **9** vs **10**; Table 1), revealed that N-methylation reduces maximal *agr* activation, which should be considered if using this strategy to stabilize thiolactone-containing AIPs. Collectively, our data show that the AgrC receptor in *L. monocytogenes* is more promiscuous with respect to structural modifications that are allowed within the agonist molecule than previously found for *S. aureus* systems. Further, **P2** is more active than **P1** and **P3** in the reporter assay, despite its very short-lived nature, and development of stabilized analogues of **P2** further supported a role of this exotail-free thiolactone as a tight-binding native AIP.

Relative quantification of *agrB* expression by qPCR revealed that **P1–P3** can all positively regulate *agr*-transcription at sub-micromolar peptide concentrations (250 nM). These experiments showed an underappreciated effect of **P1** compared to our reporter strain data, demonstrating that both **P1** and **P2** could restore *agrB* expression in the Δ*agrD* mutant to levels that were not significantly different from *agrB* expression in the untreated WT. This result also embraces the data recorded for **P1** in a different reporter strain that relies on fluorescence read-out[20]. Of the three compounds, only **P2** could restore expression of another *agr*-regulated gene, *lmo0477*, which encodes a putative secreted protein[18]. Further, the highly potent thioether analogue of **P2** (compound **10**) could increase expression of both *agrB* and *lmo0477* in the Δ*agrD* mutant to levels above the untreated WT, although the difference was only statistically significant for the latter gene. Nevertheless, the results favor **P2** as a potent regulator of the *L. monocytogenes agr* system and opens the possibility of an *agr* system employing multiple AIP variants with varying effects, potentially for adaptive regulation of *agr* activity. In that respect, the qPCR experiments further underline the different effects of **P2** and its spontaneously rearranged analogue **P3**. Our findings, in turn, raise several interesting questions regarding the potential biological significance of this transient nature of **P2**.

It has been proposed that *agr* in *L. monocytogenes* not only serves the purpose of sensing cell density[5], and differential regulation of *agr* has been shown to depend on temperature and overlap with other regulons[18]. Further, heterogeneous *agr* activity at population level was observed for several strains of *L. monocytogenes*, and it is possible that this short-lived signal may function as a single-cell autoinducer under certain conditions[29].

This work adds to the potential effects of outside stimuli on *agr* regulation in *L. monocytogenes*. The pH-dependent stability of the peptide signal (**P2**), as previously reported[21], could be speculated to provide a way to fine tune *agr* activity in response to changing pH in the environment. For example, a regulatory role for **P2** could be imagined in specific compartments, such as acidifying primary vacuoles or phagolysosomes of the host cells. These acidified compartments provide the pH for optimum activity of listeriolysin O (LLO). From the vacuole or phagolysosome, LLO helps *L. monocytogenes* escape into the neutral environment of the host cell cytosol, where LLO then rapidly denatures[30,31]. Further, *L. monocytogenes* infection relies on traversing the environments of the gastro-intestinal tract and its varying pH values[1]. It is therefore intriguing to start thinking about how such a pH-dependent peptide signal might play a role during this process, possibly even in concert with the more chemically stable peptides **P1** and **P3**.

The present work adds to a growing body of data, indicating that the prototypic and extensively studied *agr* system of *S. aureus* may not provide a fulfilling model for the understanding of *agr* in other Gram-positive bacteria. For example, a common structural feature of AIPs identified from different Gram-positive species is the lack of a peptide exotail[22,32,33], which is present in all known staphylococcal AIPs. Moreover, knowledge of a master regulatory RNA – like RNAIII in the staphylococcal *agr* systems – is missing in *L. monocytogenes*[3]. Albeit, the *L. monocytogenes agr* system has been found to downregulate the expression of the regulatory RNA, LhrA[7], that otherwise suppresses expression of the proposed virulence factor chitinase, ChiA[34,35]. Thus, providing a link between *agr* and regulation of virulence in *L. monocytogenes*.

This research provides essential insight into the peptide-mediated cell-to-cell communication of *L. monocytogenes* and provides tool compounds for future studies of the biological importance of *agr* regulation in this bacterium. More generally, several other bacteria also produce exotail-free AIPs, and we hope that the chemical framework provided in this study can help guide further investigation of their respective physiological regulation through cell-to-cell communication[22,32,33].

## Methods

### Assay protocol for monitoring of S→N acyl shift
A microcentrifugal tube (1.5 mL) containing a solution of phosphate buffer (pH = 7, 100 mM) and MeCN (1:1) was preheated to 37 °C. Assays were initiated (*t* = 0 min) by the addition of 1 µL of peptide DMSO stock solution per 100 µL of buffer–MeCN solution (100 µM final peptide conc.) with subsequent mixing using an orbital shaker. Reactions were incubated at 37 °C under constant stirring and samples were taken at relevant time points and mixed (20:1, v/v) with a solution of TFA–water (1:1, v/v) to prevent further rearrangement. Samples were then analyzed by UPLC (gradient: 5–95% of eluent B over 5 min) and ratios between exotail-free thiolactone and homodetic peptide was determined by integration of the areas under the corresponding peaks at a wavelength of $\lambda$ = 215 nm. Rate constants (*k*) were calculated as the slope of plotting S→N acyl shift progression over time using GraphPad Prism 9.0 software.

### Luciferase-based assay protocol for measuring *agr* activity in *L. monocytogenes*
*L. monocytogenes* bioluminescent reporter strains (WT::P2-lux or ΔagrD::P2-lux), previously constructed by Zetzmann et al.[8] were streaked onto TSA plates containing 5 µg/mL chloramphenicol and grown at 37 °C overnight. Single colonies were used to inoculate overnight cultures in TSB/BHI containing 5 µg/mL chloramphenicol for use in the assay. To the outer wells of a white 96-well assay plate with flat clear bottom was added sterile water, to minimize evaporation from the remaining 60 wells during

incubation. Peptide stocks in DMSO (5 mM) were diluted in TSB/BHI to the appropriate concentration (10× the final assay concentration) and 20 µL was added to the respective wells (to "No peptide" control wells were added 20 µL TSB instead of diluted peptide stock unless otherwise specified). The optical density at 600 nm (OD$_{600}$) of overnight cultures was measured and cultures were diluted to an OD$_{600}$ = 0.01 in fresh TSB containing flavin mononucleotide (11.1 mg/L, final assay concentration; ~10 mg/L, ~26 µM). Subsequently, 180 µL of diluted overnight culture was added to the relevant wells of the assay plate. The assay plate was incubated in a BioTek Synergy H1 microplate reader at 37 °C with continuous double orbital shaking. Luminescence (gain = 255) and OD$_{600}$ was measured every 20 min for 14 h.

### Relative quantification of *agrB* and *lmo*0477 by quantitative real-time polymerase chain reaction (qPCR) analysis
*L. monocytogenes* EGDe and *L. monocytogenes* EGDeΔ*agrD* overnight cultures were diluted to OD$_{600}$ = 0.01 in fresh TSB medium (10 mL) and transferred to 100 mL flasks. For each strain, cultures were prepared where DMSO (50 µL) or peptides **P1**–**P3** or **10** in DMSO (50 µL, 250 nM final peptide concentration) were added and all flasks were incubated with shaking at 37 °C. Cells were harvested on ice upon entry into stationary phase (6 h). Disruption of cell pellets were performed with a FastPrep instrument (M.P. Biomedicals), followed by RNA extraction, using the QIAGEN RNeasy kit, according to the manufacturer's instructions. RNA samples were subsequently treated with DNase, using the TURBO DNA-*free* kit (Invitrogen), and subjected to assessment of RNA purity and concentration by NanoDrop. The integrity was assessed by agarose gel electrophoresis. Complimentary DNA (cDNA) was generated with the high-capacity cDNA reverse transcription kit (Applied Biosystems), according to the manufacturer's instruction. FastStart essential DNA green master mix (Roche) was used together with the cDNA samples to conduct fluorescence-based qPCR in a LightCycler96 qPCR instrument (Roche). The following primers were used for *agrB*: CCTTTGTCAGAAAGAATGGC and CGATACCGTATACGAGAGC. The following primers were used for *lmo*0477: GCTATGATGAAAGAATAGACTTACC and TCTCACCTTCTGTTTGTCC. For amplification of the reference gene (*gyrA*) the following primers were used: CCTAGACTATGCGATGAGTG and CGAGCCGATTTTTTATAGGC. The obtained Cq values were analyzed by the 2$^{\Delta\Delta Cq}$ method for relative quantification of gene expression and compared by ordinary one-way ANOVA test, using GraphPad Prism software (10.1.1).

### General procedures for automated solid-phase peptide synthesis (SPPS)
The following Fmoc-protected amino acids (Fmoc-AA-OH) were used for the automated synthesis of peptides: Fmoc-Ala-OH, Fmoc-Met-OH, Fmoc-Phe-OH, and Fmoc-Val-OH. Automated peptide synthesis was carried out on a Biotage SyroWave™ synthesizer using iterative coupling and deprotection steps on 3-4-amino-(methylamino)benzoic acid (MeDbz)-Gly-ChemMatrix resin (0.45 mmol/g) or pre-loaded 2-chlorotrityl chloride (Cl-Trt)-polystyrene resin (0.7–0.9 mmol/g). *Loading of an amino acid residue to MeDbz-resin.* The coupling reaction was performed for 90 min at room temperature with short vortexing intervals (10 s) using stock solutions of Fmoc-AA-OH in DMF (5 equivalents to the resin loading, 0.5 M), 2-(1*H*-7-azabenzotriazol-1-yl)-1,1,3,3-tetramethyluronoium hexafluorophosphate (HATU) in DMF (4.9 equivalents, 0.5 M) and *i*-Pr$_2$NEt in NMP (10 equivalents, 2.0 M) at a final concentration of 0.2 M for Fmoc-AA-OH. The coupling was followed by washing of the resin with DMF (5 × 1 min) and the procedure was repeated to achieve a double coupling. *Standard coupling of an amino acid residue.* The coupling reaction was performed for 40 min at room temperature with short vortexing intervals (10 s) using stock solutions of Fmoc-AA-OH in DMF (5 equivalents to the resin loading, 0.5 M), 2-(1H-7-benzotriazol-1-yl)-1,1,3,3-tetramethyluronoium hexafluorophosphate (HBTU) in DMF (4.9 equivalents, 0.5 M) and

$i$-Pr$_2$NEt in NMP (10 equivalents, 2.0 M) at a final concentration of 0.2 M for Fmoc-AA-OH. The coupling reaction was followed by washing of the resin with DMF (5 × 1 min) and the procedure was repeated to achieve a double coupling. Fmoc removal was performed in two stages: (1) piperidine in DMF (2:3, v/v) for 3 min and (2) piperidine in DMF (1:4, v/v) for 12 min with short vortexing intervals (10 s). The deprotection was followed by washing of the resin with DMF (3 × 1 min), CH$_2$Cl$_2$ (1 × 1 min) and DMF (3 × 1 min).

### General procedure for manual coupling step of N-terminal amino acids

The following amino acids were used in the manual coupling step as $N$-terminal amino acid: $N,N$-dimethyl-Cys(Trt)-OH (**S2**), Fmoc-Dap(Alloc)-OH, $N,N$-dimethyl-Dap(Fmoc)-OH (**S16**), Boc-Cys(S$t$-Bu)-OH, $N,N$-dimethyl-Cys[Val(Fmoc)]-OH (**S35**), and Fmoc-Ser(TBDMS)-OH. After automated peptide elongation, the resin was transferred into a polypropylene syringe equipped with a fritted disk using CH$_2$Cl$_2$ and the resin was then washed with DMF (3 × 1 min). The coupling reaction was performed using amino acid (2 equivalents to the resin loading), HATU (2 equivalents), and $i$-Pr$_2$NEt (4 equivalents) in DMF (final concentration = 0.08 M for Fmoc-AA-OH) at room temperature under light shaking. After 2 h, the coupling mixture was removed by suction and the resin was washed with DMF (3 × 1 min) and CH$_2$Cl$_2$ (3 × 1 min).

### General procedure for the synthesis of N,N-dimethyl-thiolactone peptides

*N-acyl-benzimidazolinone (Nbz) formation:* After completed peptide elongation on 20.0 μmol MeDbz-Gly-ChemMatrix resin (see Supplementary Fig. 7), the peptidyl-MeDbz resin (1 equivalent) was washed with CH$_2$Cl$_2$ (5 × 1 min). A solution of 4-nitrophenyl-chloroformate (5 equivalents) in CH$_2$Cl$_2$ (conc: 0.1 M) was added to the resin and the suspension was agitated for 30 min. The resin was then washed with CH$_2$Cl$_2$ (2 × 1 min) and the procedure was repeated. The resin was then washed with CH$_2$Cl$_2$ (3 × 1 min) and DMF (3 × 1 min) and a solution of $i$-Pr$_2$NEt (25 equivalents, 0.5 M) in DMF was added to the resin. After 15 min, the resin was washed with DMF (3 × 1 min) and the procedure was repeated. The peptidyl-MeNbz-resin was then washed with DMF (3 × 1 min), $i$-Pr$_2$NEt in DMF (5%, v/v) (3 × 1 min), DMF (3 × 1 min), MeOH (3 × 1 min), and CH$_2$Cl$_2$ (3 × 1 min) and dried under high vacuum. *On-resin cleavage-inducing cyclization:* The dried peptidyl-MeNbz-Gly-ChemMatrix resin (20.0 μmol, 1 equivalent; Supplementary Fig. 7) was treated with a deprotection cocktail (2.0 mL, TFA–$i$-Pr$_3$SiH–water, 94:3:3, v/v/v) for 1 h and the TFA cocktail was subsequently removed from the resin. The resin was washed with CH$_2$Cl$_2$ (3 × 1 min), DMF (3 × 1 min), and CH$_2$Cl$_2$ (3 × 1 min) and dried under suction for several minutes. Cyclization buffer (5.0 mL, phosphate buffer (0.2 M, pH 6.8)–MeCN, 1:1, v/v) was added to the resin (final concentration = 4.0 mM) and the suspension was agitated at 50 °C. After 2 h, the solution was collected, and the resin was rinsed with fresh cyclization buffer. The combined peptide-containing washings were pooled and purified by preparative HPLC to afford the $N,N$-dimethyl-thiolactone peptides **1–5** after lyophilization.

### N,N-Dimethyl-P2 (1)

Peptide **1** was synthesized on MeDbz-resin on 20 μmol scale using the general procedures for automated SPPS, manual coupling of $N,N$-dimethyl-Cys(Trt)-OH (**S2**) and the synthesis of $N,N$-dimethyl-thiolactone peptides. Preparative RP-HPLC purification (5–95% B over 30 min) afforded the trifluoroacetate salt of peptide **1** as a white solid (4.7 mg, 6.1 μmol, 31%). UPLC purity (λ 215 nm): 95%; $^1$H NMR (600 MHz, DMSO-$d_6$): δ 10.51 (s, 1H), 8.92 (d, $J$ = 7.7 Hz, 1H), 8.77 (d, $J$ = 9.1 Hz, 1H), 8.35 (d, $J$ = 9.9 Hz, 1H), 7.31–7.15 (m, 11H), 4.69–4.62 (m, 1H), 4.60 (dd, $J$ = 9.9, 4.4 Hz, 1H), 4.39 (ddd, $J$ = 12.3, 8.3, 4.3 Hz, 1H), 3.76 (td, $J$ = 8.2, 6.1 Hz, 1H), 3.72–3.68 (m, 1H), 3.47 (dd, $J$ = 12.9, 4.3 Hz, 1H), 3.30–3.15 (m, 2H), 3.10 (dd, $J$ = 13.6, 5.1 Hz, 1H), 2.82 (dd, $J$ = 13.6, 10.9 Hz, 1H), 2.77 (dd, $J$ = 12.9, 11.7 Hz, 1H),

2.69 (s, 3H), 2.43 (pd, $J$ = 6.9, 4.5 Hz, 1H), 2.34 (s, 3H), 2.18–2.10 (m, 2H), 1.96 (s, 3H), 1.94–1.81 (m, 2H), 0.99 (d, $J$ = 6.9 Hz, 3H), 0.95 (d, $J$ = 6.8 Hz, 3H); UPLC-MS ($m/z$): [M + H]$^+$ calcd. for C$_{33}$H$_{46}$N$_5$O$_5$S$_2^+$, 656.29; found 656.35.

### N,N-Dimethyl-P2 F2A (2)

Peptide **2** was synthesized on MeDbz-resin on 20 μmol scale using the general procedures for automated SPPS, manual coupling of $N,N$-dimethyl-Cys(Trt)-OH (**S2**) and the synthesis of $N,N$-dimethyl-thiolactone peptides. Preparative RP-HPLC purification (5–95% B over 30 min) afforded the trifluoroacetate salt of peptide **2** as a white solid (4.3 mg, 6.2 μmol, 31%). UPLC purity (λ 215 nm): 97%; $^1$H NMR (600 MHz, DMSO-$d_6$): δ 10.49 (s, 1H), 8.64–8.38 (m, 3H), 8.24 (d, $J$ = 9.9 Hz, 1H), 7.57–7.04 (m, 5H), 4.58 (dd, $J$ = 9.9, 4.2 Hz, 1H), 4.39 (p, $J$ = 7.4 Hz, 1H), 4.34–4.21 (m, 1H), 3.88–3.75 (m, 1H), 3.67 (q, $J$ = 7.3 Hz, 1H), 3.48 (dd, $J$ = 13.0, 4.5 Hz, 1H), 3.29–3.16 (m, 2H), 2.92–2.84 (m, 1H), 2.79 (s, 6H), 2.41 (pt, $J$ = 6.6, 3.3 Hz, 1H), 2.24 (t, $J$ = 7.5 Hz, 2H), 2.01–1.95 (m, 4H), 1.92–1.83 (m, 1H), 1.24 (d, $J$ = 7.2 Hz, 3H), 0.94 (t, $J$ = 6.7 Hz, 6H); UPLC-MS ($m/z$): [M + H]$^+$ calcd. for C$_{27}$H$_{42}$N$_5$O$_5$S$_2^+$, 580.26; found 580.21.

### N,N-Dimethyl-P2 M3A (3)

Peptide **3** was synthesized on MeDbz-resin on 20 μmol scale using the general procedures for automated SPPS, manual coupling of $N,N$-dimethyl-Cys(Trt)-OH (**S2**) and the synthesis of $N,N$-dimethyl-thiolactone peptides. Preparative RP-HPLC purification (5–95% B over 30 min) afforded the trifluoroacetate salt of peptide **3** as a white solid (3.8 mg, 5.4 μmol, 27%). UPLC purity (λ 215 nm): 98%; $^1$H NMR (600 MHz, DMSO-$d_6$): δ 10.30 (s, 1H), 8.74 (s, 1H), 8.49 (d, $J$ = 9.3 Hz, 1H), 8.41 (d, $J$ = 8.0 Hz, 1H), 8.24 (d, $J$ = 10.0 Hz, 1H), 7.38–7.06 (m, 10H), 4.69–4.64 (m, 1H), 4.62 (dd, $J$ = 10.0, 3.9 Hz, 1H), 4.22 (q, $J$ = 7.8 Hz, 1H), 3.79–3.71 (m, 1H), 3.71–3.63 (m, 1H), 3.46–3.41 (m, 1H), 3.26 (d, $J$ = 7.8 Hz, 2H), 3.02 (dd, $J$ = 13.6, 4.5 Hz, 1H), 2.78–2.66 (m, 2H), 2.65–2.22 (m, 10H), 1.26 (d, $J$ = 6.9 Hz, 3H), 0.98 (d, $J$ = 6.9 Hz, 3H), 0.94 (d, $J$ = 6.9 Hz, 3H); UPLC-MS ($m/z$): [M + H]$^+$ calcd. for C$_{31}$H$_{42}$N$_5$O$_5$S$^+$, 596.29; found 596.23.

### N,N-Dimethyl-P2 F4A (4)

Peptide **4** was synthesized on MeDbz-resin on 20 μmol scale using the general procedures for automated SPPS, manual coupling of $N,N$-dimethyl-Cys(Trt)-OH (**S2**) and the synthesis of $N,N$-dimethyl-thiolactone peptides. Preparative RP-HPLC purification (5–95% B over 30 min) afforded the trifluoroacetate salt of peptide **4** as a white solid (1.5 mg, 2.2 μmol, 11%). UPLC purity (λ 215 nm): 91%; $^1$H NMR (600 MHz, DMSO-$d_6$): δ 10.36 (s, 1H), 9.03 (s, 1H), 8.56 (d, $J$ = 9.5 Hz, 1H), 8.42 (d, $J$ = 7.8 Hz, 1H), 8.09 (d, $J$ = 9.9 Hz, 1H), 7.33–7.16 (m, 5H), 4.74 (ddd, $J$ = 11.2, 9.5, 4.6 Hz, 1H), 4.53 (dd, $J$ = 9.9, 4.2 Hz, 1H), 4.19 (p, $J$ = 7.4 Hz, 1H), 3.86–3.79 (m, 1H), 3.73 (s, 1H), 3.43 (dd, $J$ = 12.9, 4.6 Hz, 1H), 3.09 (dd, $J$ = 13.6, 4.6 Hz, 1H), 2.76–2.70 m, 2H, 2.47–2.29 (m, 3H), 2.19–2.09 (m, 1H), 2.06–2.00 (m, 4H), 1.43 (d, $J$ = 7.3 Hz, 3H), 0.96 (d, $J$ = 6.9 Hz, 3H), 0.91 (d, $J$ = 6.8 Hz, 3H); UPLC-MS ($m/z$): [M + H]$^+$ calcd. for C$_{27}$H$_{42}$N$_5$O$_5$S$_2^+$, 580.26; found 580.24.

### N,N-Dimethyl-P2 V5A (5)

Peptide **5** was synthesized on MeDbz-resin on 20 μmol scale using the general procedures for automated SPPS, manual coupling of $N,N$-dimethyl-Cys(Trt)-OH (**S2**) and the synthesis of $N,N$-dimethyl-thiolactone peptides. Preparative RP-HPLC purification (5–95% B over 30 min) afforded the trifluoroacetate salt of peptide **5** as a white solid (5.1 mg, 6.9 μmol, 35%). UPLC purity (λ 215 nm): 98%; $^1$H NMR (600 MHz, DMSO-$d_6$): δ 9.00 (d, $J$ = 7.6 Hz, 1H), 8.94 (d, $J$ = 9.3 Hz, 1H), 8.40 (d, $J$ = 9.1 Hz, 1H), 8.33 (d, $J$ = 8.4 Hz, 1H), 7.38–7.05 (m, 10H), 4.68 (td, $J$ = 9.9, 5.4 Hz, 1H), 4.61–4.53 (m, 1H), 4.40 (ddd, $J$ = 12.1, 8.4, 4.4 Hz, 1H), 3.71–3.64 (m, 1H), 3.62 (dd, $J$ = 11.6, 4.3 Hz, 1H), 3.44–3.38 (m, 1H), 3.18 (dd, $J$ = 13.6, 4.4 Hz, 1H), 3.09–3.00 (m, 2H), 2.84–2.71 (m, 2H), 2.14 (t, $J$ = 7.3 Hz, 2H), 2.01–1.95 (m, 4H), 1.95–1.87 (m, 1H), 1.37 (d, $J$ = 7.0 Hz, 3H); UPLC-MS ($m/z$): [M + H]$^+$ calcd. for C$_{31}$H$_{42}$N$_5$O$_5$S$_2^+$, 628.26; found 628.24.

### N,N-Dimethyl-P2 C1Dap (6)

(Supplementary Fig. 11). The protected linear peptide **S18** was synthesized in 40.0 μmol scale on Cl-Trt polystyrene resin preloaded with Fmoc-Val-OH **S17** (0.90 mmol/g) using the general procedures for automated SPPS and manual coupling of N,N-dimethyl-Dap(Fmoc)-OH (**S16**). After completed peptide elongation, a solution of piperidine in DMF (2.0 mL, 1:4, v/v) was added to the resin, which was agitated at room temperature for 15 min. The resin was then washed with DMF (3 × 1 min) and $CH_2Cl_2$ (3 × 1 min) and dried under suction for 15 min. The dried resin was treated with a cleavage cocktail (3.0 mL, TFA–i-Pr$_3$SiH–water, 95:2.5:2.5, v/v/v) for 30 min at room temperature. The cleavage solution was removed from the resin, collected and the resin rinsed with TFA (1.0 mL). The combined cleavage solution and the rinsing solution were concentrated under a stream of nitrogen and the cleaved peptide **S19** was triturated in cold Et$_2$O (10 mL) and pelleted by centrifugation. The crude peptide **S19** (27 mg, 30.5 μmol) was obtained as the TFA salt and used without further purification.

The linear peptide **S19** (27 mg, 30.5 μmol, 1 equiv) was dissolved in anhydrous DMF (3.0 mL) under nitrogen atmosphere and added dropwise to a solution of benzotriazol-1-yl-oxy-tris-pyrrolidinophosphonium hexafluorophosphate (PyBOP) (15.9 mg, 30.5 μmol, 1 equiv) and i-Pr$_2$NEt (21.2 μL, 122 μmol, 4 equiv) in anhydrous DMF (28.0 mL). The reaction mixture was stirred for 16 h at room temperature and after full consumption of **S19** was confirmed by UPLC-MS, the reaction was reduced to dryness under reduced pressure. The remaining residue was purified by preparative RP-HPLC (20–70% B over 30 min) to afford the trifluoroacetate salt of peptide **6** as a white solid (9.0 mg, 12.0 μmol, 39% from peptide **S19**). UPLC purity (λ 215 nm): 98%*; $^1$H NMR (600 MHz, DMSO-$d_6$): δ 10.03 (s, 1H), 9.20 (d, J = 6.8 Hz, 1H), 8.56 (d, J = 8.0 Hz, 1H), 8.01 (d, J = 7.4 Hz, 1H), 7.59 (d, J = 9.8 Hz, 1H), 7.31–7.17 (m, 11H), 4.67–4.60 (m, 1H), 4.29 (dd, J = 9.8, 5.1 Hz, 1H), 4.01 (ddd, J = 11.5, 7.3, 4.1 Hz, 1H), 3.94–3.86 (m, 2H), 3.55–3.50 (m, 1H), 3.38–3.19 (m, 3H), 3.08 (dd, J = 13.8, 5.3 Hz, 1H), 2.80–2.71 (m, 2H), 2.71–2.55 (broad signal for $(CH_3)_2N^+$, 6H), 2.38–2.30 (m, 1H), 2.29–2.21 (m, 2H), 2.02–1.98 (m, 4H), 1.97–1.88 (m, 1H), 0.96 (d, J = 6.9 Hz, 3H), 0.93 (d, J = 6.8 Hz, 3H). *The final peptide sample contains tri(pyrrolidin-1-yl)phosphine oxide as non-UV active (215–280 nm) impurity (340 μg in 9.0 mg, ~10 mol%) from the PyBOP cyclization step, which was inseparable by RP-HPLC; UPLC-MS (m/z): [M + H]$^+$ calcd. for $C_{33}H_{47}N_6O_5S^+$, 639.33; found 639.30.

### P2 C1Dap (7)

(Supplementary Fig. 12). The protected linear peptide **S20** was synthesized in 80.0 μmol scale on Cl-Trt polystyrene resin preloaded with Fmoc-Val-OH **S17** (0.90 mmol/g) using the general procedures for automated SPPS and manual coupling of Fmoc-Dap(Alloc)-OH. After completed peptide elongation, a solution of di-tert-butyl dicarbonate (Boc$_2$O) (349 mg, 1.60 mmol, 20 equiv) and i-Pr$_2$NEt (0.42 mL, 2.40 mmol, 30 equiv) in DMF (5.0 mL) was added to the resin **S20** and it was agitated at room temperature. After 1 h, the solution was removed and the Boc-protected resin **S21** was washed with DMF (3 × 1 min) and $CH_2Cl_2$ (3 × 1 min) and dried under high vacuum. The resin **S21** was swelled in anhydrous $CH_2Cl_2$ (3.0 mL) for 15 min and a solution of Pd(PPh$_3$)$_4$ (18.5 mg, 16.0 μmol, 0.2 equiv), Me$_2$NH·BH$_3$ (23.6 mg, 0.40 mmol, 5 equiv) in anhydrous $CH_2Cl_2$ (3.0 mL) was then added. The resin was agitated at room temperature for 15 min and the solution was removed by suction before addition of a fresh solution of Pd(PPh$_3$)$_4$–Me$_2$NH·BH$_3$–$CH_2Cl_2$. After another 15 min, the solution was removed by suction and the resin washed with DMF (3 × 1 min) and $CH_2Cl_2$ (3 × 1 min) and dried overnight under vacuum. The dried resin was treated with hexafluoro-isopropyl alcohol (HFIP) in $CH_2Cl_2$ (5.0 mL, 1:4, v/v) for 30 min at room temperature. The cleavage solution was collected and a fresh HFIP–$CH_2Cl_2$ solution was added to the resin. After 30 min, the cleavage solution was collected, and the resin was rinsed with $CH_2Cl_2$ (5.0 mL). The combined cleavage and the rinsing solution were concentrated under reduced pressure to yield the crude peptide **S22**, which was purified by preparative RP-HPLC (5–95% B over 30 min) to afford the trifluoroacetate salt of peptide **S22** as a white solid (24 mg, 28.5 μmol). The

partially protected peptide **S22** (15.0 mg, 17.8 μmol) was dissolved in anhydrous DMF (3.0 mL) under nitrogen atmosphere and added dropwise to a solution of HATU (6.77 mg, 17.8 μmol, 1 equiv) and i-Pr$_2$NEt (12.4 μL, 71.2 μmol, 4 equiv) in anhydrous DMF (15 mL). The reaction mixture was stirred for 16 h at room temperature and after full consumption of **S22** was confirmed by UPLC-MS, the solvent was removed under reduced pressure. The resulting residue was treated with a solution of TFA in $CH_2Cl_2$ (4.0 mL, 1:1, v/v) for 1 h and subsequently concentrated under a stream of nitrogen and purified by preparative RP-HPLC (5–95% B over 30 min) to afford the trifluoroacetate salt of peptide **7** as a white solid (3.2 mg, 4.4 μmol, 25% from peptide **S22**). UPLC purity (λ 215 nm): 98%; $^1$H NMR (600 MHz, DMSO-$d_6$): δ 9.13 (d, J = 6.7 Hz, 1H), 8.47 (d, J = 7.0 Hz, 1H), 8.15 (s, 3H), 7.98 (d, J = 7.3 Hz, 1H), 7.61 (d, J = 9.6 Hz, 1H), 7.36–7.16 (m, 10H), 7.10 (t, J = 6.1 Hz, 1H), 4.56–4.48 (m, 1H), 4.25 (dd, J = 9.6, 5.4 Hz, 1H), 4.05–3.98 (m, 1H), 3.93–3.88 (m, 1H), 3.87–3.79 (m, 1H), 3.55–3.49 (m, 1H), 3.39–3.33 (m, 1H), 3.31–3.20 (m, 2H), 3.02 (dd, J = 14.0, 5.9 Hz, 1H), 2.84 (dd, J = 14.0, 9.3 Hz, 1H), 2.32–2.24 (m, 1H), 2.21–2.09 (m, 2H), 2.03–1.85 (m, 5H), 0.96–0.91 (m, 6H); UPLC-MS (m/z): [M + H]$^+$ calcd. for $C_{31}H_{43}N_6O_5S^+$, 611.30; found 611.24.

### P2 C1Ser (8)

(Supplementary Fig. 14). The partially protected linear peptide **S24** was synthesized in 80.0 μmol scale on Cl-Trt polystyrene resin preloaded with Fmoc-Phe-OH **S23** (0.87 mmol/g) using the general procedures for automated SPPS and manual coupling of Fmoc-Ser(TBDMS)-OH. After completed peptide elongation, a solution of Boc$_2$O (349 mg, 1.60 mmol, 20 equiv), i-Pr$_2$NEt (0.42 mL, 2.40 mmol, 30 equiv) in DMF (5.0 mL) was added the peptidyl-resin **S24** with free N-terminal amine and the resin was agitated at room temperature. After 1 h, the solution was removed and the Boc-protected peptidyl-resin **S25** was washed with DMF (3 × 1 min) and $CH_2Cl_2$ (3 × 1 min) and dried under high vacuum. The resin **S25** was swelled in anhydrous THF (5.0 mL) for 15 min and a solution of tetrabutylammonium fluoride (TBAF) in THF (1.0 M) (0.80 mL, 0.80 mmol, 10 equiv) in anhydrous THF (4.2 mL) was added to the resin, which was agitated at room temperature for 1 h. Then, the TBAF solution was removed by suction and the resin treated with a fresh TBAF–THF solution. After 1 h, the TBAF solution was removed by suction and the resin washed with DMF (3 × 1 min) and $CH_2Cl_2$ (3 × 1 min) and dried overnight under vacuum. On-resin esterification was performed according to a previously published protocol[36]. The resin was swelled in anhydrous $CH_2Cl_2$ (3.0 mL) for 15 min and subsequently a solution of Fmoc-Val-OH (136 mg, 0.40 mmol, 5 equiv), N,N'-diisopropylcarbodiimide (DIC) (75.2 μL, 0.48 mmol, 6 equiv) and N-methylimidazole (NMI) (17.2 μL, 0.22 mmol, 5.4 equiv) in anhydrous $CH_2Cl_2$ (2.0 mL) was added. The resin was agitated at room temperature for 2 h and then washed with anhydrous $CH_2Cl_2$ (3 × 1 min). A fresh Fmoc-Val-OH–DIC–NMI solution was added to the resin and after 2 h of incubation, the resin was washed with DMF (3 × 1 min), $CH_2Cl_2$ (3 × 1 min), and DMF (3 × 1 min).

Fmoc-removal from resin **S26** was performed by treatment with a solution of 1,8- biazabicyclo[5.4.0]undec-7-ene (DBU) in DMF (2.0 mL, 1:99, v/v) (8 × 30 s). The resin was then washed with DMF (3 × 1 min) and $CH_2Cl_2$ (3 × 1 min) and dried under suction for 15 min. The dried resin was treated with a solution of HFIP in $CH_2Cl_2$ (5.0 mL, 1:4, v/v) for 30 min at room temperature. The cleavage solution was collected and a fresh HFIP–$CH_2Cl_2$ solution was added to the resin. After 30 min the second cleavage solution was collected, and the resin was rinsed with $CH_2Cl_2$ (5.0 mL). The combined cleavage and the rinsing solutions were evaporated to dryness under reduced pressure to yield the crude peptide **S27**, which was purified by preparative RP-HPLC (5–95% B over 30 min) to afford the trifluoroacetate salt of peptide **S27** as a white solid (30 mg, 35.6 μmol). The partially protected peptide **S27** (15.0 mg, 17.8 μmol) was dissolved in anhydrous DMF (3.0 mL) under nitrogen atmosphere and added dropwise to a solution of HATU (6.77 mg, 17.8 μmol, 1.00 equiv) and i-Pr$_2$NEt (12.4 μL, 71.2 μmol, 4.00 equiv) in anhydrous DMF (15 mL). The reaction mixture was stirred for 16 h at room temperature and after full consumption

of **S27** was confirmed by UPLC-MS, the solvent was removed under reduced pressure. The resulting residue was treated with a solution of TFA in $CH_2Cl_2$ (4.0 mL, 1:1, v/v) for 1 h and was then concentrated under a stream of nitrogen and purified by preparative RP-HPLC (5–95% B over 30 min) to afford the trifluoroacetate salt of peptide **8** as a white solid (6.8 mg, 9.4 µmol, 53% from peptide **S27**). UPLC purity ($\lambda$ 215 nm): 97%; $^1$H NMR (600 MHz, DMSO-$d_6$): δ 9.12 (d, $J$ = 6.9 Hz, 1H), 8.86 (d, $J$ = 4.5 Hz, 1H), 8.19 (s, 3H), 7.32 (d, $J$ = 7.3 Hz, 1H), 7.30–7.25 (m, 7H), 7.25–7.18 (m, 2H), 7.14–7.09 (m, 2H), 4.58 (dd, $J$ = 12.2, 3.1 Hz, 1H), 4.48 (dd, $J$ = 9.6, 4.8 Hz, 1H), 4.37–4.29 (m, 2H), 4.24 (t, $J$ = 2.5 Hz, 1H), 4.02 (ddd, $J$ = 11.3, 7.1, 4.3 Hz, 1H), 3.50 (ddd, $J$ = 10.4, 6.8, 3.7 Hz, 1H), 3.41–3.34 (m, 2H), 2.93 (dd, $J$ = 14.1, 8.6 Hz, 1H), 2.86 (dd, $J$ = 14.1, 6.8 Hz, 1H), 2.26–2.09 (m, 2H), 2.02–1.97 (m, 1H), 1.96 (s, 3H), 1.91–1.83 (m, 1H), 0.89 (d, $J$ = 6.8 Hz, 3H), 0.85 (d, $J$ = 6.8 Hz, 3H); UPLC-MS ($m/z$): $[M + H]^+$ calcd. for $C_{31}H_{42}N_5O_6S^+$, 612.29; found 612.21.

### *N,N*-Dimethyl-P2 thioether (9)

(Supplementary Fig. 16). The protected linear peptide **S36** was synthesized in 25.0 µmol scale on Cl-Trt polystyrene resin preloaded with Fmoc-Phe-OH **S23** (0.87 mmol/g) using the general procedures for automated SPPS and manual coupling of *N,N*-dimethyl-Cys[Val(Fmoc)]-OH (**S35**). After completed peptide elongation, a solution of piperidine in DMF (2.0 mL, 1:4, v/v) was added to the resin, which was agitated at room temperature for 15 min. The resin was then washed with DMF (3 × 1 min) and $CH_2Cl_2$ (3 × 1 min) and dried under suction for 15 min. The dried resin was treated with a cleavage cocktail of TFA–$i$-Pr$_3$SiH–water (3.0 mL, 95:2.5:2.5, v/v/v) for 30 min at room temperature. The cleavage solution was collected, and the resin rinsed with TFA (1.0 mL). The combined cleavage and rinsing solutions were concentrated under a stream of nitrogen and the cleaved peptide **S37** was triturated in cold Et$_2$O (10 mL) and pelleted by centrifugation. The crude peptide **S37** was obtained as TFA salt and used without further purification.

The linear peptide **S37** was dissolved in anhydrous DMF (3.0 mL) under nitrogen atmosphere and added dropwise to a solution of PyBOP (13.0 mg, 25.0 µmol, 1 equiv) and $i$-Pr$_2$NEt (17.4 µL, 100 µmol, 4 equiv) in anhydrous DMF (22 mL). The reaction mixture was stirred for 16 h at room temperature and after full consumption of **S37** was confirmed by UPLC-MS, the solvent was removed under reduced pressure. The resulting residue was purified by preparative RP-HPLC (20–70% B over 30 min) and two diastereoisomers of **9** were isolated due to partial loss epimerization of building block **S35**. The major isomer was concluded to correspond to the desired peptide and was obtained as the trifluoroacetate salt of peptide **9** as a white solid (4.3 mg, 5.7 µmol, 23% based on resin). UPLC purity ($\lambda$ 215 nm): 95%; $^1$H NMR (600 MHz, DMSO-$d_6$): δ 10.01 (s, 1H), 9.00 (d, $J$ = 6.8 Hz, 1H), 8.87 (d, $J$ = 9.1 Hz, 1H), 8.17 (d, $J$ = 8.1 Hz, 1H), 7.31–7.23 (m, 7H), 7.23–7.16 (m, 4H), 4.86–4.79 (m, 1H), 4.06 (ddd, $J$ = 12.1, 8.8, 4.2 Hz, 1H), 3.87–3.83 (m, 1H), 3.77–3.69 (m, 1H), 3.54 (ddd, $J$ = 9.0, 7.1, 5.5 Hz, 1H), 3.25–3.08 (m, 5H), 2.75 (dd, $J$ = 13.7, 11.1 Hz, 1H), 2.62 (dd, $J$ = 11.0, 3.8 Hz, 1H), 2.53–2.47 (m, 2H), 2.28–2.23 (m, 2H), 2.08–1.93 (m, 5H), 1.83–1.74 (m, $J$ = 6.7 Hz, 1H), 0.95–0.90 (m, 6H); UPLC-MS ($m/z$): $[M + H]^+$ calcd. for $C_{33}H_{48}N_5O_4S_2^+$, 642.31; found 642.36.

### P2 thioether (10)

(Supplementary Fig. 17). The fully protected linear peptide **S38** was synthesized in 80.0 µmol scale on Cl-Trt polystyrene resin preloaded with Fmoc-Phe-OH **S23** (0.87 mmol/g) using the general procedures for automated SPPS and manual coupling of Boc-Cys($St$-Bu)-OH. After completed peptide elongation, a solution of *N*-methylmorpholine (NMM) (44.2 µL, 0.40 mmol, final concentration = 0.1 M) in a mixture of β-mercaptoethanol–DMF (4.0 mL, 1:4, v/v) was added to the resin **S38** and the resin was agitated for 16 h at room temperature. Then, the thiol-containing solution was removed by suction and the resin **S39** was washed with DMF (3 × 1 min) and $CH_2Cl_2$ (3 × 1 min) and dried overnight under vacuum. The resin **S39** was then swelled in anhydrous DMF (3.0 mL) for 15 min. In a

separate flask, a suspension of $CsCO_3$ (130 mg, 0.40 mmol, 5 equiv) in anhydrous DMF (5.0 mL) was sonicated for 15 min before Fmoc-Val-iodide (**S30**) (174 mg, 0.40 mmol, 5 equiv) was added. The alkylation mixture was then added to the resin **S39** and the resin was agitated at room temperature for 16 h. The solubles were removed by suction and the alkylated resin **S40** was washed with DMF (3 × 1 min), MeOH (3 × 1 min), and DMF (3 × 1 min). A solution of piperidine in DMF (3.0 mL, 1:4, v/v) was added to the resin and it was agitated at room temperature for 15 min. The resin was then washed with DMF (3 × 1 min) and $CH_2Cl_2$ (3 × 1 min) and dried under suction for 15 min. The dried resin was treated with a solution of HFIP in $CH_2Cl_2$ (5.0 mL, 1:4, v/v) for 30 min at room temperature. The cleavage solution was collected and a fresh HFIP–$CH_2Cl_2$ solution was added to the resin. After 30 min the cleavage solution was collected and, the resin was rinsed with $CH_2Cl_2$ (5.0 mL). The combined cleavage and rinsing solutions were evaporated to dryness under reduced pressure to yield the crude peptide **S41**, which was purified by preparative RP-HPLC (5–95% B over 30 min) to afford the trifluoroacetate salt of peptide **S41** as a white solid (11.2 mg, 13.2 µmol). The partially protected peptide **S41** (11.2 mg, 13.2 µmol) was dissolved in anhydrous DMF (2.0 mL) under nitrogen atmosphere and added dropwise to a solution of PyBOP (6.87 mg, 13.2 µmol, 1 equiv) and $i$-Pr$_2$NEt (9.20 µL, 52.8 µmol, 4 equiv) in anhydrous DMF (11.0 mL). The reaction mixture was stirred for 16 h at room temperature and after full consumption of **S41** was confirmed by UPLC-MS, the solvent was removed under reduced pressure. The resulting residue was treated with a solution of TFA in $CH_2Cl_2$ (3.0 mL, 1:1, v/v) for 1 h and subsequently concentrated under a stream of nitrogen and purified by preparative RP-HPLC (5–95% B over 30 min) to afford the trifluoroacetate salt of peptide **10** as a white solid (4.0 mg, 5.5 µmol, 42% from peptide **S41**). UPLC purity ($\lambda$ 215 nm): 97%; $^1$H NMR (600 MHz, DMSO-$d_6$): δ 8.95 (d, $J$ = 7.1 Hz, 1H), 8.86 (d, $J$ = 5.6 Hz, 1H), 8.15 (s, 3H), 7.85 (d, $J$ = 7.5 Hz, 1H), 7.32–7.20 (m, 8H), 7.20–7.16 (m, 3H), 4.54–4.41 (m, 1H), 4.08–4.00 (m, 1H), 3.98–3.92 (m, 1H), 3.58–3.53 (m, 1H), 3.50–3.44 (m, 1H), 3.33–3.23 (m, 2H), 3.07 (dd, $J$ = 14.0, 3.8 Hz, 1H), 2.99–2.86 (m, 3H), 2.83 (dd, $J$ = 14.0, 7.6 Hz, 1H), 2.78 (dd, $J$ = 12.4, 3.6 Hz, 1H), 2.05–1.96 (m, 2H), 1.97–1.90 (m, 4H), 1.90–1.81 (m, 2H), 0.89 (d, $J$ = 6.7 Hz, 3H), 0.87 (d, $J$ = 6.8 Hz, 3H); UPLC-MS ($m/z$): $[M + H]^+$ calcd. for $C_{31}H_{44}N_5O_4S_2^+$, 614.28; found 614.25.

**Statistics and reproducibility**. GraphPad Prism 9.0 software was used to calculate the rate constants for S→N acyl shift progression over time by linear regression. Averaging of three individual biological replicates (n = 3) of luciferase reporter strain assay data and calculation of standard error of the mean (SEM) was performed using GraphPad Prism software (version 9.0 and newer). Cq values obtained from qPCR experiments, performed in biological triplicate (n = 3), were analyzed by the $2^{\Delta\Delta Cq}$ method using Microsoft Office Excel 2016 and further evaluation of statistical significance between treatments and control was peformed by ordinary one-way ANOVA testing with multiple comparisons using Graphpad Prism software (10.1.1). All source data has been compiled and is available in Supplementary Data files 1–5. The number of biological replicates (n) is also stated in each individual figure caption.

### Reporting summary

Further information on research design is available in the Nature Portfolio Reporting Summary linked to this article.

## Data availability

The authors declare that the data supporting the findings of this study are available within the paper and its supplementary information and supplementary data files.

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

## Acknowledgements

We gratefully acknowledge the Independent Research Fund Denmark −Natural Sciences (Grant No. 0135-00427B; C.A.O.) and the LEO Foundation Open Competition Grant program (LF-OC-20-000517; M.S.B. and H.I., LF-OC-19-000039; C.A.O., and LF-OC-21-000901; C.A.O.) for financial support. We thank Prof. Christian Riedel (University of Ulm) for generously providing the bacterial strains used in the study.

## Author contributions

Benjamin S. Bejder: conceptualization, investigation, formal analysis, methodology, visualization, writing–original draft, writing–review, editing; Fabrizio Monda: investigation, formal analysis, methodology, writing–review & editing; Bengt H. Gless: conceptualization, investigation, formal analysis, methodology, supervision, visualization, writing–review & editing; Martin S. Bojer: investigation, formal analysis, methodology, supervision,

writing–review & editing; Hanne Ingmer: funding acquisition, supervision, writing–review & editing; Christian A. Olsen: conceptualization, formal analysis, funding acquisition, project administration, resources, supervision, writing-original draft, writing–review & editing.

## Competing interests

The authors declare no competing interests.
