## [Peer Review File · Communications Biology]

Reviewers' comments:

Reviewer #1 (Remarks to the Author):

In this manuscript, Bejder et al. present a collection of chemically modified analogues to the presumed *Listeria monocytogenes* quorum sensing autoinducing peptide (AIP). Their modifications allowed for stabilization of short-lived pentameric thiolactone-containing AIPs and study of their effects on the Agr quorum sensing system of *L. monocytogenes*. While this N,N-dimethyl P2 molecule was slightly less effective at activating the Agr system compared to native activation, for the first time this stabilization allowed the authors to study the role of each individual amino acid of the AIP molecule on quorum sensing signaling by alanine substitutions. Furthermore, alternative methods of stabilization of the AIP produced molecules that were more potent quorum sensing activators compared to endogenous AIP molecules. This work demonstrated that the AgrC has broader recognition of pentameric AIP peptides than previously appreciated. Together, the authors have created a suite of molecules that more closely resemble the native, short-lived AIP and will allow researchers to better probe how *L. monocytogenes* relies on quorum sensing to cause infection.

Specific comments:

1. Supplementary figure 1: Legend describes a third graph (RLU/OD600) that is not shown.
2. Figure 3a (and all similar figures): The authors show in Figure S2, P1 Luminescence panel differences in the “No peptide” condition RLU profile depending on the presence of DMSO. Please clarify whether the “No peptide” condition contains an equal amount of DMSO compared to all of the wells receiving peptide in DMSO.
3. The authors mention that a 1 μM concentration of AIP has been reported for *S. aureus* overnight cultures and admit that given its transient nature, P2 concentrations are not likely to approach these levels. However, induction with P2 in the $\Delta\text{agrD}::\text{P2-lux}$ strain required $\sim 5 \mu\text{M}$ concentrations while induction with P3 required $\sim 50 \mu\text{M}$ concentrations. Is this discrepancy an issue with the sensitivity of the assay or does the concentration of AIP reach significantly higher levels in *L. monocytogenes* cultures? Is a lower concentration of P2 required at acidic conditions due to the longer lifetime?

Reviewer #2 (Remarks to the Author):

The work of Bejder et al provides interesting and novel insights into the chemical nature of AIPs in *Listeria monocytogenes* by first identifying P2 as an inducer of agr promoter activity and then developing peptides with a longer half life which will improve the ability to study these AIPs. The work also carries out alanine scanning of the AIPs in order to identify essential amino acids for activity and in so doing demonstrated the importance of the phenylalanine positions.

Main comments

Overall the Introduction and results are well written but I feel the Discussion section requires more attention in order to contextualise the data. In my opinion the opening statement of the Discussion does not give enough detail about what is already known in *S. aureus* versus *Listeria* and the Discussion would benefit from a re-iteration of the main findings of the current work (some of this text is actually in the final section of the Results section). I am a bit unclear as to what is meant by Line 248 – “A compartmentalized regulatory role for P2 could be imagined”.

Overall I feel the discussion could be improved greatly in order to more simply present to the reader what the current study shows, how it advances the state of knowledge for *Listeria* biology and how it compares to *S. aureus*. Perhaps finishing with speculation as to the role of P2 versus P1 and P3 in the infectious cycle as written (although this section could be clarified in my opinion)?

Compound 2 seems to inhibit the *agr* promoter in the wild-type (which makes its own peptide but not in the Δ *agrD* mutant (which does not) (Fig 3d) – is there an explanation for this?

Minor Comments

Fig 4 - figure is more pixelated than the others

Line 31 – suggest changing ‘loudly suggests’ to something like ‘strongly suggests’

Line 62 – suggest changing ‘has been identified’ to ‘was identified’

Reviewer #3 (Remarks to the Author):

Summary:

In their manuscript "A short-lived peptide signal regulates cell-to-cell communication in *Listeria monocytogenes*" (Manuscript Number: COMMSBIO-23-3405-T), Bejder et al. investigate the native AIP of *L. monocytogenes* as well as different chemically stabilized variants using luciferase reporters. The data suggests that the most active for of the native AIP is instable at pH7.

Overall impression:

The study is scientifically sound and experimental approaches are appropriate with necessary controls in place. The experimental data support the conclusions. However, the work is a bit preliminary and one-dimensional as only luciferase reporter assays are performed to assess activity of the different peptides. The study would certainly benefit from additional biological data e.g. by biofilm assays or invasion assays after growth in the presence of activating and inhibitory peptides. Also, the hypothesis of a role of the active AIP in compartment sensing could be tested by analyzing stability of the peptide under conditions mimicking the gastrointestinal tract and the phagolysosome.

Specific comments:

1. Line 29 “stabilized”, correct typo

2. Line 89: change to "...that if P2 were to act as..."
3. Line 92-104: Please explain how P2 can induce a signal in the lux-reporters around 240min of the assay, when half-life is 1.3 min and 1 μM of (active) P2 is reduced to $<1\text{nM}$ within 13min.
4. Discussion: The authors conclude that AgrC is more promiscuous to structural analogues of P2. Please discuss the biological implications. *L. monocytogenes* thrives in densely populated habitats. How likely is it that there are peptides (or agonists of AgrC) produced by other bacteria or simply present in nature?
5. Discussion: The authors discuss a potential role of the AIP in compartment sensing in the gastrointestinal tract or inside vacuoles in host cells. The phagolysosome but also primary vacuoles of non-phagocytic cells are compartments with specific conditions namely low pH. Similarly, the gut has a pH below 7. How stable is P2 in acidic environments. It is suggested to provide additional experiments on P2 decay under conditions mimicking the phagolysosome.
6. Discussion: Are there any other implications of the synthetic analogues besides the importance for structure-function relationships e.g. as antagonists to inhibit virulence?
7. The study is a bit one-dimensional as it is limited to luciferase reporters. It is suggested to provide additional biological data e.g. by testing the effect of activating/inhibitory variants on biofilm formation or host cell invasion. This is important to see if the effects on activation/inhibition of the agr promoter actually translates to biological outcomes.

Reviewer #1 (Remarks to the Author):

In this manuscript, Bejder et al. present a collection of chemically modified analogues to the presumed *Listeria monocytogenes* quorum sensing autoinducing peptide (AIP). Their modifications allowed for stabilization of short-lived pentameric thiolactone-containing AIPs and study of their effects on the Agr quorum sensing system of *L. monocytogenes*. While this N,N-dimethyl P2 molecule was slightly less effective at activating the Agr system compared to native activation, for the first time this stabilization allowed the authors to study the role of each individual amino acid of the AIP molecule on quorum sensing signaling by alanine substitutions. Furthermore, alternative methods of stabilization of the AIP produced molecules that were more potent quorum sensing activators compared to endogenous AIP molecules. This work demonstrated that the AgrC has broader recognition of pentameric AIP peptides than previously appreciated. Together, the authors have created a suite of molecules that more closely resemble the native, short-lived AIP and will allow researchers to better probe how *L. monocytogenes* relies on quorum sensing to cause infection.

Specific comments:

1. Supplementary figure 1: Legend describes a third graph (RLU/OD600) that is not shown.

We thank the reviewer for noticing this mistake. We had inadvertently forgotten to modify the caption after redesigning the figure. The caption is now fitting the graphical material.

2. Figure 3a (and all similar figures): The authors show in Figure S2, P1 Luminescence panel differences in the “No peptide” condition RLU profile depending on the presence of DMSO. Please clarify whether the “No peptide” condition contains an equal amount of DMSO compared to all of the wells receiving peptide in DMSO.

This is a good point raised by the reviewer. The “No peptide” condition does generally not contain DMSO aside from comparative experiments denoted as “no peptide + 1% DMSO”, which have been added in Fig. 2 and in Supplementary Fig. S2-5. We deemed the effect of DMSO to be of minor importance based on these data points; especially because the amount is rapidly diluted much lower levels for the peptide concentrations of interest. We have now altered the experimental procedures and captions to clarify which conditions were applied in each assay.

3. The authors mention that a 1 μM concentration of AIP has been reported for *S. aureus* overnight cultures and admit that given its transient nature, P2 concentrations are not likely to approach these levels. However, induction with P2 in the $\Delta\text{agrD}::\text{P2-lux}$ strain required $\sim 5 \mu\text{M}$ concentrations while induction with P3 required $\sim 50 \mu\text{M}$ concentrations. Is this discrepancy an issue with the sensitivity of the assay or does the concentration of AIP reach significantly higher levels in *L. monocytogenes* cultures? Is a lower concentration of P2 required at acidic conditions due to the longer lifetime?

This is also a good question, which indeed was the central question that led us to perform the current study. As also discussed in our manuscript, there is certainly a discrepancy between the amount of AIP (in the form of **P3**) that we can isolate from spent medium vs. the concentration needed for activation of QS. It may be that **P3** forms disulfides as dimers or with cysteine residues in the medium, to complicate its isolation, but nevertheless we find it unlikely that its concentration would reach substantially higher levels than for *S. aureus* AIPs, based on our previously reported isolation data. Certainly, **P2** is more stable at lower pH but directly testing its effect at lower pH was prohibited by sluggish cell growth. As also discussed below, addressing a comment from reviewer 3, our current hypothesis is that the **P2** may have a very slow off rate from the receptor, in turn being protected from rearrangement when bound to the AgrC. The very high potency recorded for our minimally modified analog of **P2**, which cannot undergo rearrangement, can be seen as support of tight binding of **P2** as well. We have attempted to more clearly discuss these issues in the revised manuscript.

Reviewer #2 (Remarks to the Author):

The work of Bejder et al provides interesting and novel insights into the chemical nature of AIPs in *Listeria monocytogenes* by first identifying P2 as an inducer of agr promoter activity and then developing peptides with a longer half life which will improve the ability to study these AIPs. The work also carries out alanine scanning of the AIPs in order to identify essential amino acids for activity and in so doing demonstrated the importance of the phenylalanine positions.

Main comments

Overall the Introduction and results are well written but I feel the Discussion section requires more attention in order to contextualise the data. In my opinion the opening statement of the Discussion does not give enough detail about what is already known in *S. aureus* versus *Listeria* and the Discussion would benefit from **a re-iteration of the main findings of the current work (some of this text is actually in the final section of the Results section).**

This text has been moved and modified to fit into the Discussion.

I am a bit unclear as to what is meant by Line 248 – “A compartmentalized regulatory role for P2 could be imagined”.

Overall I feel the discussion could be improved greatly in order to more simply present to the reader what the current study shows, **how it advances the state of knowledge for *Listeria* biology and how it compares to *S. aureus*. Perhaps finishing with speculation as to the role of P2 versus P1 and P3 in the infectious cycle as written (although this section could be clarified in my opinion)?**

We thank the reviewer for the suggestion and have rewritten the Discussion substantially to more clearly frame the discoveries of the current study.

Regarding Line 248 specifically, we have rephrased this entire paragraph.

Compound 2 seems to inhibit the agr promoter in the wild-type (which makes its own peptide but not in the deltaAgrD mutant (which does not) (Fig 3d) – is there an explanation for this?

As mentioned by the reviewer, the compound acts as an inhibitor of *agr* in wild-type, which is corroborated by the lack of activation in the deltaAgrD mutant, where no activation occurs unless the applied compound is an agonist.

Minor Comments

Fig 4 - figure is more pixelated than the others

The figure has been replaced with a higher resolution version.

Line 31 – suggest changing ‘loudly suggests’ to something like ‘strongly suggests’ Line 62 – suggest changing ‘has been identified’ to ‘was identified’

Thank you, we have made both changes as suggested.

Reviewer #3 (Remarks to the Author):

Summary:

In their manuscript "A short-lived peptide signal regulates cell-to-cell communication in *Listeria monocytogenes*" (Manuscript Number: COMMSBIO-23-3405-T), Bejder et al. investigate the native AIP of *L. monocytogenes* as well as different chemically stabilized variants using luciferase reporters. The data suggests that the most active for of the native AIP is instable at pH7.

Overall impression:

The study is scientifically sound and experimental approaches are appropriate with necessary

controls in place. The experimental data support the conclusions. However, the work is a bit preliminary and one-dimensional as only luciferase reporter assays are performed to assess activity of the different peptides. The study would certainly benefit from additional biological data e.g. by biofilm assays or invasion assays after growth in the presence of activating and inhibitory peptides. Also, the hypothesis of a role of the active AIP in compartment sensing could be tested by analyzing stability of the peptide under conditions mimicking the gastrointestinal tract and the phagolysosome.

Specific comments:

1. Line 29 “stabilized”, correct typo – has been corrected

2. Line 89: change to “...that if P2 were to act as...” – has been corrected

3. Line 92-104: Please explain how P2 can induce a signal in the lux-reporters around 240min of the assay, when half-life is 1.3 min and 1 μ M of (active) P2 is reduced to <1nM within 13min.

This is indeed puzzling, but similar results have been reported by Blackwell and coworkers (*Biochemistry* 2023), using a different reporter strain system, and we can only speculate as to what the explanation for these observations may be. Currently, our hypothesis is that the **P2** may have a very slow off rate from the receptor, in turn being protected from rearrangement when bound to the AgrC. The very high potency recorded for our minimally modified analog of **P2**, which cannot undergo rearrangement, can be seen as support of tight binding of **P2** as well. We have attempted to more clearly discuss these issues in the revised manuscript.

4. Discussion: The authors conclude that AgrC is more promiscuous to structural analogues of P2. Please discuss the biological implications. *L. monocytogenes* thrives in densely populated habitats. How likely is it that there are peptides (or agonists of AgrC) produced by other bacteria or simply present in nature?

This is an interesting question but also one that we would be reluctant to speculate about. Bacterial crosstalk is common within staphylococci but has only been sporadically tested for *Listeria* so far. What we meant by promiscuity, was in relation to structural modification of the native AIP(s) that still allow for agonistic AIP activity, which is also corroborated by the SAR performed by Blackwell and coworkers (*Biochemistry* 2023). We have made this part of our discussion more clear in the revised manuscript.

5. Discussion: The authors discuss a potential role of the AIP in compartment sensing in the gastrointestinal tract or inside vacuoles in host cells. The phagolysosome but also primary vacuoles of non-phagocytic cells are compartments with specific conditions namely low pH. Similarly, the gut has a pH below 7. How stable is P2 in acidic environments. It is suggested to provide additional experiments on P2 decay under conditions mimicking the phagolysosome.

The stability of **P2** at different pH values has been investigated in detail and reported in our initial publication. We have now made this clear in the discussion.

6. Discussion: Are there any other implications of the synthetic analogues besides the importance for structure-function relationships e.g. as antagonists to inhibit virulence?

We agree that development of antagonists is another important aspect of performing SAR studies, and we do actually discover a molecule with inhibitory function in compound **2**. We have elaborated slightly on the discussion of findings, including with respect to the findings reported by Blackwell and coworkers in their recent publication (*Biochemistry* 2023).

7. The study is a bit one-dimensional as it is limited to luciferase reporters. It is suggested to provide additional biological data e.g. by testing the effect of activating/inhibitory variants on biofilm formation or host cell invasion. This is important to see if the effects on activation/inhibition of the agr promoter actually translates to biological outcomes.

We agree with the reviewer on this point, especially in light of the manuscript appearing in the meantime by Blackwell and coworkers (*Biochemistry* **2023**), using a different reporter assay, which produces slightly different results. To address these issues, we performed two lines of experiments: 1) biofilm formation assays and 2) qPCR experiments.

1) For the biofilm formation assays, we compared the *L. monocytogenes* EGD WT and $\Delta agrD$ mutant in LB medium, following the procedure from the manuscript by Blackwell and coworkers (*Biochemistry* **2023**). We could not reproduce their results, observing no difference between WT and mutant, and therefore did not pursue the testing of any compounds. Consulting the literature on *agr* and its effect on biofilm formation in *L. monocytogenes*, we found another example where one group was able to show a significant difference between EGD WT and $\Delta agrD$ in a static biofilm assay (Lee et al., *MicrobiologyOpen* **2020**), while another group could not obtain such results (Zetzmann et al., *MicrobiologyOpen* **2019**). Both groups used diluted BHI medium (10%) at 37 °C. Furthermore, it appears that the biofilm deficiency often ascribed to *agr* deficiency, is highly dependent on the assay conditions, with one study finding a $\Delta agrA$ mutant producing more biofilm compared with the WT at 37 °C, while producing less at 25 °C (Garmyn et al., *PLOS One* **2012**).

2) The qPCR experiments proved to be an important addition to the study. They demonstrated that indeed our thioether analog of **P2** (compound **10**) was a potent agonist, but also revealed that all the naturally occurring AIPs **P1-P3** are active at lower concentrations than assessed by the reporter strain assay. We have included a new results section detailing these important new data as well as a discussion of their implications.

REVIEWERS' COMMENTS:

Reviewer #1 (Remarks to the Author):

My previous points of criticism/suggestions have been addressed satisfactorily.

Reviewer #2 (Remarks to the Author):

The revised manuscript by Bejder et al. is an informative study which significantly extends our knowledge of the agr system in *Listeria monocytogenes*. The revisions have been completed and have improved the manuscript.

Reviewer #3 (Remarks to the Author):

The authors have convincingly addressed all comments and have provided a substantially revised manuscript. The revised manuscript also contains qPCR data on expression of agrB and lmo0477 further supporting their results in by a reporter-independent approach. I only have one small additional comment. The authors use lux reporters in the EGDe WT and delta-agrD background but do not provide information how these strains were constructed. Please either include a small section on strain construction (in materials and methods or the supplementary information) or a reference to the original publication of these strains.